# SEVIR : A Storm Event Imagery Dataset for Deep Learning Applications in Radar and Satellite Meteorology

**Mark S. Veillette**[*]
MIT Lincoln Laboratory
Lexington, MA 02420
mark.veillette@ll.mit.edu

**Siddharth Samsi**[*]
MIT Lincoln Laboratory
Lexington, MA 02420
sid@mit.edu

**Christopher J. Mattioli**
Amazon Web Services [†]
cmmattio@amazon.com

## Abstract

Modern deep learning approaches have shown promising results in meteorological applications like precipitation nowcasting, synthetic radar generation, front detection and several others. In order to effectively train and validate these complex algorithms, large and diverse datasets containing high-resolution imagery are required. Petabytes of weather data, such as from the Geostationary Environmental Satellite System (GOES) and the Next-Generation Radar (NEXRAD) system, are available to the public; however, the size and complexity of these datasets is a hindrance to developing and training deep models. To help address this problem, we introduce the Storm EVent ImagRy (SEVIR) dataset - a single, rich dataset that combines spatially and temporally aligned data from multiple sensors, along with baseline implementations of deep learning models and evaluation metrics, to accelerate new algorithmic innovations. SEVIR is an annotated, curated and spatio-temporally aligned dataset containing over 10,000 weather events that each consist of 384 km x 384 km image sequences spanning 4 hours of time. Images in SEVIR were sampled and aligned across five different data types: three channels (C02, C09, C13) from the GOES-16 advanced baseline imager, NEXRAD vertically integrated liquid mosaics, and GOES-16 Geostationary Lightning Mapper (GLM) flashes. Many events in SEVIR were selected and matched to the NOAA Storm Events database so that additional descriptive information such as storm impacts and storm descriptions can be linked to the rich imagery provided by the sensors. We describe the data collection methodology and illustrate the applications of this dataset with two examples of deep learning in meteorology: precipitation nowcasting and synthetic weather radar generation. In addition, we also describe a set of metrics that can be used to evaluate the outputs of these models. As of this writing, the SEVIR dataset can be downloaded from https://registry.opendata.aws/sevir/. Baseline implementations of selected applications are also available at https://github.com/MIT-AI-Accelerator/neurips-2020-sevir.

## 1 Introduction

The Earth's weather is continuously monitored by sensors that collect terabytes of data every day. Over the US, satellite observations provided by GOES-R series satellites (GOES-16 & GOES-17), and weather radar data provided by the national network of WSR-88D (NEXRAD) radars are two

---

[*]Equal contributors to this work
[†]Contributions to this work were made prior to affiliation with AWS

major sources of weather sensing used by forecasters, decision makers, and the general public. Archives of these two sensing modalities make up petabytes of data that include images of clouds in both visible and infrared, depictions of precipitation intensity, and detection of lightning. Recently, there has been a great deal of work to use deep learning to better leverage these rich data sources, specifically for applications like short term weather forecasting [30, 20, 6], synthetic weather radar for areas lacking traditional weather radar [27], improved data assimilation for improved numerical weather prediction [8], and many others. Furthermore, access to datasets like GOES & NEXRAD is becoming easier as cloud services such as Google Earth Engine [12], Amazon Open Data Registry [1], IBM's PAIRS [3] and others provide access to Earth system datasets.

While this deluge of weather data is generally a boon for machine learning, there still remains significant challenges for researchers trying to work in this area. The totality of datasets like GOES & NEXRAD is simply too large to be used directly for model training, and so significant down sampling of the data is required. This down sampling is a non-trivial step. Naive sampling techniques, like fixed region sampling or uniform random sampling, can potentially lead to datasets that under-represent severe weather or extreme events which are relatively rare. Moreover, combining data across different sensing modalities like radar and satellite requires significant compute resources to properly align these data sources on a common grid. This complexity often leads to researchers curating their own specialized datasets, which are usually not distributed or shared. Aside from a few exceptions (e.g. [2]), the community lacks a sufficient number of common "machine-learning ready" datasets in the area of meteorology that are useful for validating and benchmarking new capabilities.

To help address these issues, we present the Storm EVent ImagRy dataset (SEVIR), a dataset designed for advancing machine learning for meteorology. SEVIR contains image sequences for over 10,000 weather events that cover 384 km x 384 km patches and span 4 hours. Images in SEVIR were sampled and aligned across 5 different sensing modalities: three channels (C02, C09, C13) from the GOES-16 advanced baseline imager (ABI) [22], NEXRAD derived vertically integrated liquid (VIL) mosaics created by the FAA's NextGenWeather processor Testbed [4], and GOES-16 Geostationary Lightning Mapper (GLM) flashes [11]. Events in SEVIR were carefully sampled to ensure the dataset contains relevant severe storm cases. The main contributions of our work are summarized below:

- Publicly available terabyte-sized SEVIR dataset of 10,000 weather events aligned across 5 imaging modalities.
- Detailed overview of two machine learning applications that can be studied using SEVIR.
- Source code for data readers, baseline model implementations, metrics, loss functions and trained models for Nowcast and Synthetic Weather Radar applications.

## 2   SEVIR: Storm Event Imagery Dataset

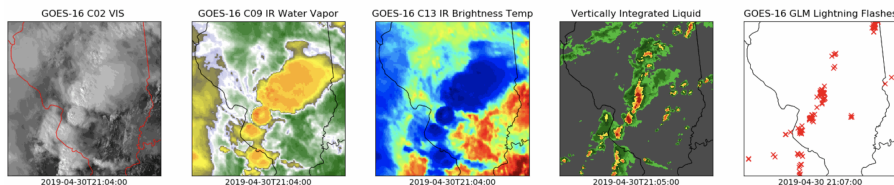

Figure 1: The Storm EVent ImagRy (SEVIR) dataset contains over 10,000 spatially and temporally aligned sequences across the five image types.

SEVIR is a collection of temporally and spatially aligned image sequences depicting weather events captured over the contiguous US (CONUS) by GOES-16 satellite and the mosaic of NEXRAD radars. Figure 1 shows a set of frames taken from a SEVIR event. SEVIR contains five image types: GOES-16 0.6 $\mu m$ visible satellite channel (`vis`), 6.9 $\mu m$ and 10.7 $\mu m$ infrared channels (`ir069`, `ir107`), a radar mosaic of vertically integrated liquid (`vil`), and total lightning flashes collected by the GOES-16 geostationary lightning mapper (GLM) (`lght`). See Table 1 for details.

Each event in SEVIR consists of a 4-hour length sequence of images sampled in 5 minute steps. The lightning modality is the only non-image type, and is represented by a collection of GLM lightning flashes captured in the 4 hour time window. SEVIR events cover 384 km x 384 km patches sampled

Table 1: Description of sensor types in SEVIR

| Image type | Description | Spatial Resolution | Patch Size | Event Count |
|---|---|---|---|---|
| vis | Visible satellite imagery (day-time only) | 0.5 km | 768x768 | 13,403 |
| ir069 | Infrared Satellite imagery (mid-level water vapor) | 2 km | 192x192 | 13,552 |
| ir107 | Infrared Satellite imagery (clean longwave window) | 2 km | 192x192 | 13,541 |
| vil | NEXRAD radar mosaic of VIL | 1 km | 384x384 | 20,393 |
| lght | Intercloud and cloud to ground lightning events | 8 km | N/A | 15,115 |

at locations throughout the continental U.S. (CONUS). The pixel resolution in the images differ by image type, and were chosen to closely match the resolution of the original data. Since the patch dimension of 384 km is constant across sensors, the size of each image differs (as shown in Table 1).

Mathematically, SEVIR events make up an indexed family $(s, i) \rightarrow X_{s,i}$, where $s \in \mathcal{S} \equiv \{\text{vis}, \text{ir069}, \text{ir107}, \text{vil}, \text{lght}\}$ is one of the five modalities, or image types, described in Table 1, and $i \in \mathcal{I}_s$ is a alphanumeric string that uniquely identifies an event captured by one or more image types. For all modalities except lght, $X_{s,i}$ is a 3D tensor with shape $[L_s, L_s, T]$ representing sequences of gray-scale images, where $L_s$ is the patch size of modality $s$ listed in Table 1, and $T$ is the number of time steps in each event (as of this writing, $T = 49$ for all image types). For lght, $X_{s,i}$ is 2D matrix of shape $\left[N_i^{lght}, 5\right]$, with columns representing time, latitude, longitude, x-position and y-position of each lightning flash for event $i$. The number of events captured by image type $s$ is given by $N_s = |\mathcal{I}_s|$. Due to gaps in sensor coverage and availability of data archives used to make SEVIR, not all events $i$ are covered by all image types. Table 1 provides the number of events covered by each image type. Approximately 12,000 events are covered by all five image types.

The images in SEVIR are stored in HDF5 files that make up 952 GB of disk space. SEVIR also contains a catalog in CSV format containing metadata for all the events, including ids $i$, image type $s$, filename and file index pointing to $X_{s,i}$, the center time stamp of the event in UTC, corner lat/lons of the event patch, the map projection of the patch, and other identifying information. The catalog also makes it possible to augment SEVIR with additional datasets as needed.

## 2.1 SEVIR Event Selection

Events in SEVIR were selected in one of two ways - *Random selection* and *Storm event based selection*. To select random events, a set of random times was uniformly selected through the years 2017-2019. For each time, each image type was retrieved if it was available. Event centers were sampled randomly based on the VIL data, with higher probability given to pixels with higher VIL intensity. This sampling method ensured that the dataset did not oversample the "no precipitation" case. Random events in SEVIR have an id starting with R.

The storm event based selection method used the National Centers for Environmental Information (NCEI) Storm Events Database[3] to target reported cases of severe weather. This database contains the time window, location, severe weather category, descriptors of storm strength and impact, as well as narratives summarizing each event. For SEVIR, entries in the Storm Events Database between 2017 and 2019 matching category Flood, Flash Flood, Hail, Heavy Rain, Lightning, Thunderstorm Wind or Tornado were selected. These events were clustered based on time, latitude and longitude using a hierarchical clustering algorithm. The central point in each identified cluster was used to select the central time and position of a SEVIR event. Storm events in SEVIR have an id starting with S.

The left panel of Figure 2 shows the geospatial locations of all SEVIR events, separated by "Random" events and "Storm" events. The times of the events counted across the 5 image types is shown in the right panel of Figure 2. The storm events are more clustered in the summer months over the US, whereas the random events are more evenly distributed. In 2017, SEVIR only contains vil, which explains the lower number of events prior to 2018.

For storm events, the SEVIR catalog also provides the EVENT_ID, EPISODE_ID, and storm type for the event in the Storm Event database used to select the SEVIR event. By cross referencing these

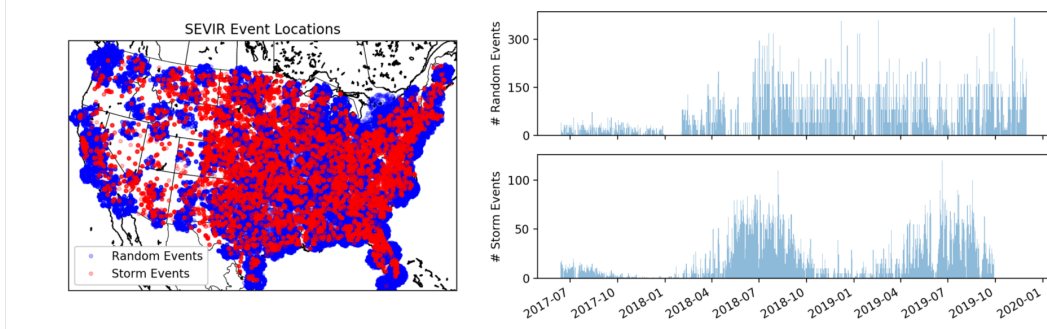

Figure 2: (Left) Location of all events collected in SEVIR. The red points represent SEVIR events that are directly linked to NCEI Storm Events. (Right) Temporal distribution of SEVIR events.

columns with the NCEI Storm Event database, several other text-based data entries can be linked to the imagery data in SEVIR. Figure 3 shows an example one such record corresponding to a SEVIR Storm Event.

| SEVIR ID | | Record in National Weather Service's Storm Event Database | | | | |
|---|---|---|---|---|---|---|

**Storm Event IDs**

S789829
**S788040**
S766963

**Event Narrative** and other table:

| Event | Scale | Deaths Direct/Indirect | Injuries Direct/Indirect | Property Damage |
|---|---|---|---|---|
| Tornado | EF2 | 0/0 | 0/0 | 200.00K |

| Episode Narrative | Event Narrative |
|---|---|
| A strong upper level disturbance interacted with a weak front and plenty of low level moisture to produce a cyclic supercell thunderstorm during the early morning hours. This cyclic supercell thunderstorm produced a tornado near Christoval and another one in Brady. | A National Weather Service team found evidence of tornado damage about 6 to 9 miles west southwest of Christoval. The strong tornadic winds snapped trunks and uprooted many Live Oak and Cedar trees. The tornado hit a metal building. The twister removed much of the roof covering, blew out a large garage door and caused steel purlins to buckle. |

**Random Event IDs**

R19113018467863
R19113018468164
R19113018468498

Figure 3: Many events in SEVIR are directly linked to the NCEI storm event database which contains data associated with SEVIR events such as storm types, storm impacts, and storm narratives.

## 3 Applications of SEVIR for problems in Weather Modeling

SEVIR was designed to address a number of problems in meteorology. Many of these problems have connections to well-known problems in computer vision and machine learning, such as:

***Future Prediction***: Given sequence $X = \{X^1, ..., X^p\}$ of consecutive frames of image type $s$, predict future frames $Y = \{Y^1, ..., Y^f\}$. This is the problem of *nowcasting*, which is a term used for short-term forecasts that typically go 1 to 2 hours into the future.

***Image-to-Image translation***: Given image types $s_i \in \mathcal{S}_{input}$, estimate unknown image type(s) $s_j \in \mathcal{S}_{output}$. For example, generate "synthetic" weather radar imagery (like that represented by `vil`), using satellite imagery and lightning detection for areas lacking ground based weather radar.

***Unsupervised optical flow***: Given a sequence of frames from image type $s$, generate a *flow field* $V \in \mathbb{R}^{L_s, L_s, 2}$ that describes motion in the image. In meteorology, this is similar to atmospheric motion estimation [9], which provides wind inputs to numerical weather prediction models.

***Classification or automatic caption generation***: Given data from multiple image types, classify storm type, and generate text based descriptions of the event similar to that shown in Figure 3.

***Super-Resolution***: Down sample images in SEVIR, and try to learn a mapping to the original higher resolution. In meteorology, this is similar to *statistical downscaling* [26].

Each problem in this (non-exhaustive) list can be studied using SEVIR. The remainder of this paper will focus on two of these examples: nowcasting and synthetic weather radar generation. Working baseline implementations and evaluations of these two capabilities are provided. Models discussed

here use SEVIR data before June 1, 2019 for training. Data during and after June 1, 2019 is the test dataset and is used only for final evaluation of the fit model.

## 3.1 Model Architecture and Loss Functions

We use variants of the U-Net architecture [19] to solve both the nowcasting and synthetic radar problems. Combined SEVIR inputs are passed through a series of 4 encoder blocks with 32, 64, 128 and 256 filters of size 3x3, followed by a bottleneck layer with 1024 2D convolutional filters with a 3x3 receptive field. The decoder portion of the network consists of 4 decoder blocks for nowcasting, and 5 in the case of synthetic radar in order to match the target resolution. The final 2D convolutional layer of the network uses a linear activation, configured with the appropriate number of outputs. (Additional details in supplementary materials). All models were implemented in TensorFlow [5] and were trained on the MIT Supercloud TX-GAIA [17, 21] system with compute nodes having two 32GB NVIDIA Volta V100 GPUs and dual 20-core Intel Xeon Gold 6248 CPUs with 384 GB of RAM. The Horovod [23] framework was used for data distributed training. Models were trained using 8 GPUs across 4 nodes.

Several variations of loss functions were tested for comparison. In both problems considered, the goal is to produce images of `vil`, thus $L^p$ loss functions like MSE or MAE are appropriate. However, it is well known that sole reliance on these loss functions can lead to poor texture [16, 31]. Therefore we also implemented advanced loss functions to test if more realistic texture can be produced in addition to spatial and temporal accuracy in the prediction. The loss functions used are as follows:

***Reconstruction Loss***: Either $L_{MSE}$ or $L_{MAE}$.

***VGG16 Content Loss***: $L_{content}(Y, \hat{Y}; \ell) = \|\phi^\ell(Y) - \phi^\ell(\hat{Y})\|_2$, where $\phi^\ell$ is the activation of the $\ell$th layer of the VGG16 network [24].

***VGG16 Style Loss***: $L_{style}(Y, \hat{Y}; \mathbf{w}) = \sum_\ell w^\ell \|G^\ell(Y) - G^\ell(\hat{Y})\|_2$, where $\mathbf{w} = \left[w^1, \ldots, w^L_{VGG16}\right]$ are weights corresponding to VGG layers, and $G^\ell$ is the gram matrix of features corresponding to the activations of layer $\ell$.

***Conditional GAN Loss***: $L_{cGAN}(X, Y, \hat{Y}) = \mathbb{E}_{x,y}\left[D(x, y)\right] + \mathbb{E}_x\left[1 - D(x, G(x))\right]$, where $G$ is the model generating a prediction given inputs $x$, $D$ is a discriminator model trying to predict if an image is either from the training set or produced by the generator, and $\mathbb{E}$ is the expectation taken w.r.t. the training data. A discriminator model similar to the pix2pix model (PatchGAN) [14] was used.

## 3.2 Evaluation Metrics

Evaluation of network performance using image quality assessment can be inherently challenging. We propose the dual approach of forecast-specific metrics as well as commonly used image quality metrics for evaluating networks trained on the SEVIR dataset. We evaluate the overall quality of the generated imagery using metrics that are common in forecast evaluation [29]. These metrics are computed by first binarizing the truth and prediction images at a set of thresholds that span the range 0 - 255. Thresholds for `vil` were chosen based on the 6 Video Integrator and Processor (VIP) intensity levels [18] which correspond to pixel values $[16, 74, 133, 160, 181, 219]$. Binarized pixels are scored as "Hits" if `prediction=truth=1`, "Misses" if `prediction=0,truth=1`, "False Alarms" if `prediction=1,truth=0` and "Correct Rejection" otherwise. The following summary statistics are computed by aggregating these counts over the test set:

Probability of detection (POD) $= \frac{\#Hits}{\#Hits + \#Misses}$ , Success Ratio (SUCR) $= \frac{\#Hits}{\#Hits + \#F.Alarm}$

Critical Success Index (CSI) $= \frac{\#Hits}{\#Hits + \#Misses + \#F.Alarms}$ , BIAS $= \frac{\#Hits + \#F.Alarms}{\#Hits + \#Misses}$

Note that POD, SUCR and CSI are equivalent to recall, precision and intersection over union (IOU), however we will keep the naming conventions used in the forecast verification literature. In addition to these metrics, the outputs of these models are also evaluated on a perceptual basis. While pixel-based metrics such as the MSE/MAE and region based metrics such as the structural similarity (SSIM) [28] can be a computationally efficient approach to comparing image quality, these do not capture the rich textural information in weather radar images. Thus, to evaluate the perceptual quality of the generated images, we propose the use of the Learned Perceptual Image Patch Similarity (LPIPS) proposed by Zhang et. al. [31], which has been shown to outperform these widely used metrics. The LPIPS metric

calculates the cosine distance between normalized network activations from deep networks such as AlexNet, SqueezeNet and VGG. In this paper we use the ImageNet [10] trained AlexNet network with fixed pre-trained network weights.

## 3.3   Radar Nowcasting

Nowcasts are high resolution, short-term (e.g. up to 2 hours) weather forecasts of radar echos, precipitation, cloud coverage or other meteorological quantities widely used in public safety, air traffic control, and many other areas that require high fidelity and rapidly updating forecasts. Previous work on deep learning for Nowcasting includes convolutional Long Short Term Memory (ConvLSTM) models [30], recurrent architectures [13] and fully convolutional networks [20, 6] for precipitation nowcasting.

We frame nowcasting as a *future prediction* task where the model input consists of 13 VIL images sampled at 5 minute intervals. The model is trained to produce the next 12 images in the sequence, corresponding to the next hour of weather. Data from SEVIR was first extracted and processed into 44,760 sequences for training and an independent set of 12,144 sequences for testing the fit model. This was done by splitting each SEVIR event into 3 input and output sequences. The model input was of size Nx384x384x13 and the output sequence was Nx384x384x12 pixels, where $N$ is the batch size.

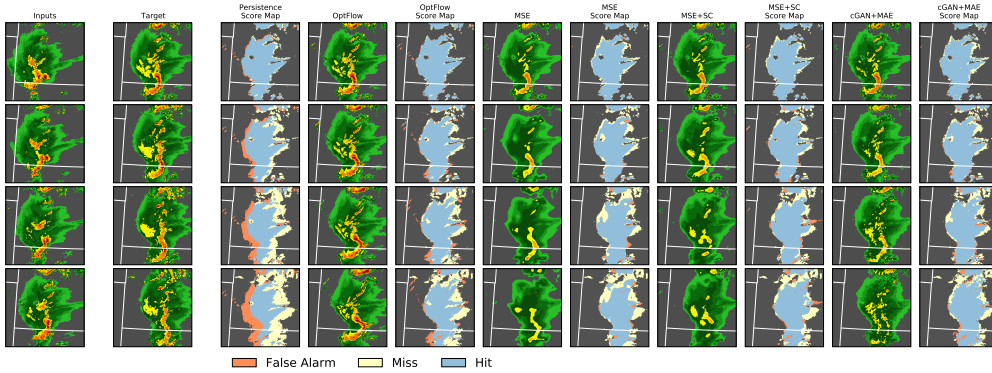

Figure 4: Nowcast output from U-Net model with different loss functions. Abbreviations used are as follows: MSE - Mean Squared Error, SC - Style and content loss, cGAN - Conditional GAN

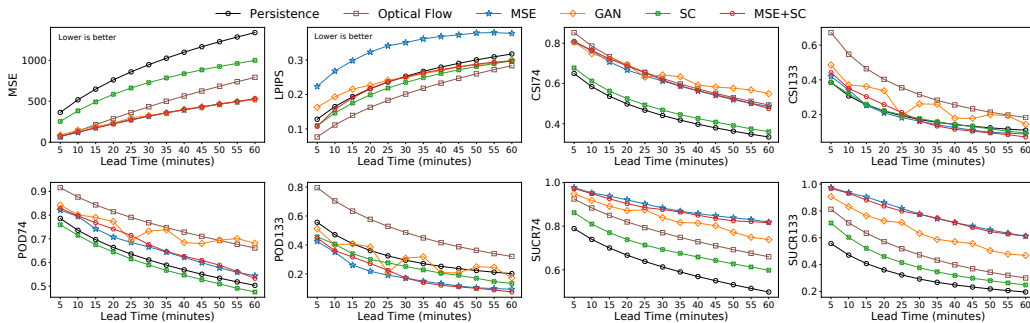

Figure 5: Trends in evaluation metrics over lead times: All approaches show improved performance over the persistence model. However, the metrics show degraded performance as the lead time increases. Numbers 74, 133 on the y-axis correspond to threshold levels described in Section 3.2

We used the U-Net architecture and four different loss functions as described in Section 3. The simplest loss function used was an $L_{MSE}$ loss which has been shown to produce forecasts that do not have sufficient detail over the entire prediction sequence [20, 25]. To improve upon this, we trained a second model using Style and Content ($SC$) loss computed from a VGG16 model trained on the ImageNet dataset as described in [15]. This model was found to generate improved textures in the predicted images, but it fails to track motion in the images. A third model that combined the $L_{MSE}$

loss and the $SC$ loss was trained to improve motion tracking as well as texture. Finally, we trained a conditional GAN (cGAN) with the same UNet as a generator and a discriminator as described in Section 3. The cGAN model was trained for 300 epochs with the Adam optimizer and a batch size of 32. All other models were trained for 100 epochs. Models that used Style and Content loss were trained with a batch size of 4 because of GPU memory constraints.

The models described above were evaluated against two common baseline models for nowcasting. The first is the *Persistence Model* which assumes that conditions do not change at successive time steps past the final input image. This evaluates whether new models outperform the most trivial model which simply repeats the last data seen. The second baseline is an optical-flow based nowcast obtained using the `rainymotion`[4] package [7]. This optical flow model estimates a dense flow field at each pixel using the last two frames in the input sequence, and creates a nowcast by advecting each pixel in the last input image by the estimated flow field using a semi-Lagrangian scheme.

Figure 4 shows sample output of an easterly moving storm (SEVIR ID S846323) comparing the persistence and optical flow benchmarks to three of the U-Net variations described above at 15, 30, 45 and 60 minute lead times. We show only the score for the persistence model since it simply repeats the last image from the input. Visually, it's apparent that the MSE-trained model washes out detail and much of the higher intensity weather for longer leads, whereas the models trained using SC or adversarial loss retains more radar-like texture at longer leads. The score columns show each of the U-Net models are able to adequately track the motion of the system, and generate fewer false alarms along the back edge of the storm compared to persistence and optical flow.

Figure 5 shows selected metrics described in 3.2 for each model. All networks optimized using reconstruction loss outperformed the benchmarks in MSE, which isn't too surprising, but noteworthy. The overall poor performance of SC only suggests some weighting of reconstruction loss is necessary, as leaving it out seems to cause forecasts to not move. MSE and MSE+SC based loss functions provide higher CSI (IOU) compared to persistence, but fail outperform optical flow. These networks seem to generate more conservative forecasts that favor higher SUCR (Precision) over POD (Recall), whereas the oppostie is true for optical flow. For moderate precipitation levels (threshold 74), the MAE+GAN yields higher CSI scores at later leads (> 25 minutes), suggesting adversarial component combined with an MAE term is effective for this problem. For higher intensity precipitation (threshold = 133), the U-Net models show lower CSI (IOU), however the GAN model is close and the gap can likely be overcome with additional hyperparameter tuning. In terms of perceptual similarity, inclusion of style or adversarial loss drastically improves the texture of the output forecast relative to an MSE loss. The LPIPs score of these neural network models are within 0.1 of both the persistence and optical flow scores, which are both expected to score well in this category as they are mainly copies of the input data and hence will retain a greater textural similarity.

### 3.4 Synthetic Weather Radar

Depictions of storms obtained from weather radar are extremely important in many areas; however, most areas of the world do not have access to ground based radar. Using SEVIR, we will train a model that creates radar-like imagery of storm depictions using only satellite and lightning as inputs. To do so, we take the set of image types $\mathcal{S}_{input} = \{\texttt{ir069,ir107,lght}\}$ as inputs to the model. We will train the U-Net described in Section 3.1 to transform these three image types into `vil`.

Data in SEVIR first needed to extracted and pre-processed prior to the U-net. The `lght` data was converted to an image by binning 5 minutes of flashes prior to `ir069` and `ir107` onto a 48 x 48 pixel grid[5]. The three input images were then resized to the `vil` size of 384x384. Each channel was normalized by subtracting their mean and dividing by their standard deviation computed over the training set. The three resized and normalized images are then passed to the U-net.

Three loss variations were tested for the synthetic radar problem: (1) MSE normalized by the variance of the target variable (2) Content loss $L_{content}$ using VGG19's `block5_conv4` layer added to MSE: $L_{content} + L_{MSE}$, and (3) Conditional adversarial loss added to MAE (similar to pix2pix): $L_{cGAN} + L_{MAE}$. For (1) and (2), training was done until the loss stopped decreasing on a validation set constructed using 20% of the training data. For (3), training was done for 200 epochs and stopped.

Table 2: Test set scores for the synthetic radar model. MSE loss leads to lowest MSE/MAE, as well as better CSI for the higher thresholds. For the low to mid thresholds, MSE+Content loss is the best performer for BIAS and CSI. Models trained with the cGAN+MAE loss generates yielded the best LPIPS score, suggesting this loss provides the best perceptual match to the target.

| Loss Metric | Thres | MSE | MSE +Content | cGAN +MAE | Metric | Thres. | MSE | MSE +Content | cGAN +MAE |
|---|---|---|---|---|---|---|---|---|---|
| MAE | - | **10.93** | 11.04 | 12.69 | | 16 | 1.1474 | **1.0787** | 0.7437 |
| | | | | | | 74 | 0.6683 | **0.7405** | 0.4507 |
| MSE | - | **466.64** | 497.26 | 738.41 | BIAS | 133 | 0.3238 | **0.4193** | 0.3106 |
| | | | | | | 160 | 0.3313 | 0.3625 | **0.3996** |
| LPIPS | - | 0.3934 | 0.6195 | **0.3498** | | 181 | 0.2889 | 0.2781 | **0.3397** |
| | | | | | | 219 | 0.0768 | 0.0799 | **0.1211** |
| | 16 | **0.8211** | 0.7868 | 0.5746 | | 16 | 0.6191 | **0.6090** | 0.4915 |
| | 74 | 0.4994 | **0.5300** | 0.3176 | | 74 | 0.4273 | **0.4378** | 0.2803 |
| POD | 133 | 0.2353 | **0.2619** | 0.1918 | CSI | 133 | 0.2162 | **0.2263** | 0.1714 |
| | 160 | **0.2496** | 0.2388 | 0.2188 | | 160 | **0.2307** | 0.2125 | 0.1853 |
| | 181 | **0.2100** | 0.1751 | 0.1602 | | 181 | **0.1947** | 0.1588 | 0.1358 |
| | 219 | **0.0432** | 0.0344 | 0.0242 | | 219 | **0.0418** | 0.0329 | 0.0221 |

Figure 6 shows the results of the three loss functions for three cases in the test set. In all cases, the

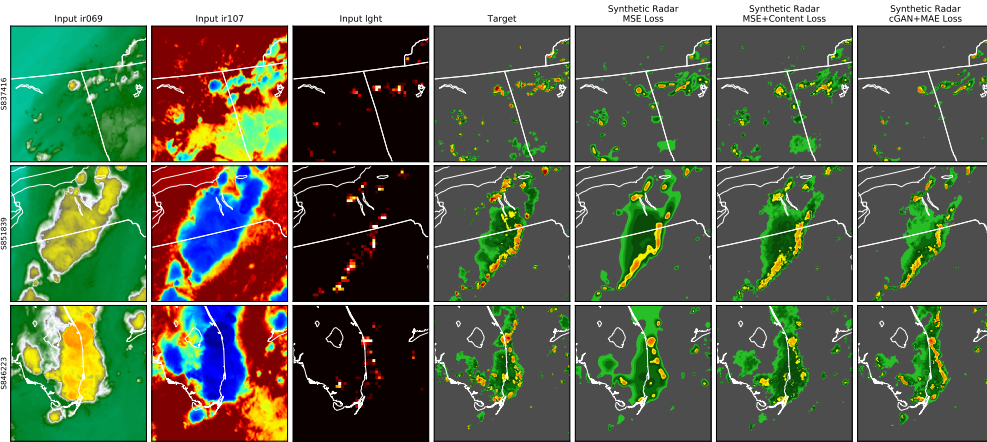

Figure 6: Three examples of the synthetic weather radar model trained using three different loss functions. MSE leads to an accurate, albeit overly smoothed, prediction. The content and adversarial losses are able to provide additional textures that are visually more similar to the target.

model was able to generate an accurate depiction of `vil`, particularly in the areas with high lightning intensity. Visually, the MSE-trained model generates overly smoothed output, whereas qualitatively, the VGG & cGAN losses provide improved texture. Table 2 shows a set of evaluation metrics applied to the test set. Unsurprisingly, the network trained with MSE loss performed the best under both MSE and MAE. The addition of content loss yielded slightly better scores in both BIAS and CSI at the 74 and 133 thresholds. The cGAN+MAE has the highest LPIPS score, suggesting this loss generated the most perceptually similar result to the target (at the expense of other metrics) as seen in Figure 6.

## 4 Conclusions

This work introduced the Storm EVent ImageRy dataset (SEVIR). SEVIR is a unique, terabyte-sized collection of over 10,000 weather events depicted by five spatially and temporally aligned sensors, including 3 channels from the GOES-16 satellite, one weather radar-derived variable (VIL), and lighting detections from the GOES GLM sensor. Each SEVIR event covers a 4 hour period of

time in 5 minute increments, sampled randomly (with over sampling of events with mid and high precipitation), and by using the NOAA Storm Event Database. SEVIR can be used to to build and evaluate models for a range of problems in meteorology. Baseline models and performance benchmarks of two problems, nowcasting and synthetic weather radar, were provided. By providing baseline implementations, evaluation metrics and a large dataset, we hope that SEVIR will spur new developments in the machine learning and meterological fields. SEVIR will continue to grow as new data becomes available and additional data modalities will be added to SEVIR as appropriate.

It is our hope that SEVIR will be used by researchers to both improve upon the models described in this work, and to solve other problems relevant to meteorology. We believe that SEVIR also presents an opportunity for researchers to address new challenges with more unintuitive data than classical image recognition/classification datasets. Additionally, this dataset presents interesting insights into the capabilities and limitations of current deep learning approaches as well as the open question of qualitative image assessment.

In addition to the applied research tasks discussed in this work, future efforts may include

- **Public Challenges** One of the main motivations in curating SEVIR was its use in future public challenges for radar and satellite meteorology. One challenge focused on nowcasting is already being planned for 2021.
- **Transfer Learning** ML researches in the earth system sciences often lack appropriate pretrained models for transfer learning approaches. SEVIR acts a benchmark dataset for providing such models and for learning representations appropriate for weather and climate forecasting. This will benefit researchers working in areas of the world with sparse weather and climate measurements.
- **Model Robustness** Weather and climate datasets are particularly susceptible to noise and sensor calibration discrepancies. SEVIR provides an idealized dataset with relatively clean data that can be artificially modified to develop robustness strategies.
- **Transparency and Explainability** Forecast models that are too opaque may limit user acceptance in an operational setting. SEVIR provides a common dataset for developing and demonstrating techniques that illuminate features that explain how certain outputs are generated.

By providing baseline implementations, evaluation metrics and a large dataset, we hope that SEVIR will spur new developments in the machine learning and meteorological fields.

## Broader Impact

This work offers a free and open dataset with the purpose of advancing machine learning applications in the area of meteorology. In addition to the dataset, we offer two benchmark problems with working implementations and evaluation metrics. These will allow other researchers to easily build off of this work to create new and enhanced capabilities. Authors do not foresee negative ethical consequences as a result of this work. A potential positive societal impact may arise from the development of models that can generate radar imagery from modalities that are not available world-wide. This could give new meteorological monitoring capabilities to underdeveloped or low resource societies.

## Acknowledgments and Disclosure of Funding

Research was sponsored by the United States Air Force Research Laboratory and was accomplished under Cooperative Agreement Number FA8750-19-2-1000. The views and conclusions contained in this document are those of the authors and should not be interpreted as representing the official policies, either expressed or implied, of the United States Air Force or the U.S. Government. The U.S. Government is authorized to reproduce and distribute reprints for Government purposes notwithstanding any copyright notation herein.

The authors acknowledge the MIT SuperCloud and Lincoln Laboratory Supercomputing Center for providing high-performance computing resources that have contributed to the research results reported in this paper.

## Footnotes

[3]https://www.ncdc.noaa.gov/stormevents/

[4]https://github.com/hydrogo/rainymotion

[5]The choice of 48 is because GLM has an approximate accuracy of 8km, and the SEVIR patches are of size 384km.

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
