[Supplementary Material]

# Supplementary Material for - SEVIR : A Storm Event Imagery Dataset for Deep Learning Applications in Radar and Satellite Meteorology

**Mark S. Veillette**[*]
MIT Lincoln Laboratory
Lexington, MA 02420
mark.veillette@ll.mit.edu

**Siddharth Samsi**[*]
MIT Lincoln Laboratory
Lexington, MA 02420
sid@ll.mit.edu

**Christopher Mattioli**
Amazon Web Services [†]
cmmattio@amazon.com

## 1 SEVIR Dataset

This section contains additional information about the SEVIR dataset, including details of the SEVIR catalog, information regarding geo-referencing events and data encoding.

### 1.1 SEVIR Catalog

The SEVIR catalog contains relevant information about each event in SEVIR. Table 1 includes a list of the catalogs columns, with a short description of each. When extracting data from SEVIR, it is helpful to first group the catalog by the `id` column. After doing so, the size of each group will represent the number of image types that are available for each event. This is useful for building training datasets that utilize multiple image types (such as the synthetic radar problem). The catalog also allows for efficient filtering of SEVIR by time, geographic location, or by statistic for more focused training and testing sets.

The grayscale images that make up SEVIR events are associated with a 384 km x 384 km patch on the Earth's surface. This patch can be exactly geo-referenced using the patch's map projection, along with a specification of the corner latitudes and longitudes. In order to perform this geo-referencing (as in done in e.g. Figure 1. of the main paper), the columns `llcrnrlat`, `llcrnrlon`, `urcrnrlat`, `urcrnrlon` and `proj` can be used. The map projection `proj` is defined as a PROJ string [1] which contains the name of the projection, Earth model, and other parameters required for defining the projection. As of this writing, all images in SEVIR use a Lambert azimuthal equal-area projection centered over the US, which was chosen to minimize distortion of the data.

### 1.2 Performance of HDF5

SEVIR data is available as a set of HDF large files. A commonly used approach in model training is to organize data (especially images) into directories that correspond to class labels. Other approaches include combining data into TFRecord files (compatible with TensorFlow) or HDF5 files. We chose HDF5 because of the availability of open source libraries for reading this data format in a variety of languages such as python, MATLAB, C/C++, JAVA, Fortran, etc. While it's possible to stream data from SEVIR for model fitting, randomized reads from HDF5 files can be slow. In addition, HDF5-specific choices (such as the chuck size), made during file creation can also affect the speed of file reads. A comprehensive analysis of HDF5 File I/O is beyond the scope of this work, but based on our experiments, it is recommended that when using SEVIR, the data first be read into one or more interim files where the data is shuffled a-priori and written to file sequentially. This removes the need

---

[*]Equal contributors to this work
[†]Contributions to this work were made prior to affiliation with AWS

Table 1: Contents of SEVIR catalog

| Column | Description |
|---|---|
| id | Unique id given to each event in SEVIR. |
| file_name | Name of the HDF5 file containing the image data |
| file_index | File index within file_name where the data is located |
| img_type | Image or sensor type |
| time_utc | UTC Timestamp of the event, which typically corresponds to the middle frame in the event |
| minute_offsets | Colon separated values denoting time offset in minutes of each frame relative to time_utc |
| episode_id | Storm Event EPISODE_ID associated to the SEVIR event (NOAA Storm Events only) |
| event_id | Storm Event EVENT_ID associated to the SEVIR event (NOAA Storm Events only) |
| llcrnrlat | Latitude of the lower left corner |
| llcrnrlon | Longitude of the lower left corner |
| urcrnrlat | Latitude of the upper right corner |
| urcrnrlon | Longitude of the upper right corner |
| proj | Proj software library style string describing the map projection underlying images in the event |
| size_x | X Size of the images in pixels |
| size_y | Y Size of the images in pixels |
| height_m | X Size of the images in meters |
| width_m | Y Size of the images in meters |
| data_min | Minimum data value across all frames of the event |
| data_max | Maximum data value across all frames of the event |
| pct_missing | Percentage of missing values across all the frames |

Table 2: SEVIR linear scaling factors

| Type | SCALING_FACTOR | Decoded units |
|---|---|---|
| vis | 1e-4 | Reflectance factor |
| ir069 | 1e-2 | Degrees C |
| ir107 | 1e-2 | Degrees C |

for data randomization during training and also ensures that each data access results in a sequential read from the file, resulting in improved read performance during fitting.

## 1.3 Data encoding

For efficient storage, data is saved in HDF5 files as an integer type. Depending on the sensor type, these integers can be decoded into floating type so they represent the actual values captured by each sensor type. This decoding is done using either a simple linear scaling, or using an exponential transformation, as described below.

The satellite images (vis, ir069 and ir107) use a linear scaling to encode their values. To convert the integer data stored in the file to floating types, apply the formula

$$\text{decoded\_data} = \text{encoded\_data} * \texttt{SCALING\_FACTOR} \tag{1}$$

where SCALING_FACTOR is provided in Table 2, along with the units of the decoded data. Missing pixels for the satellite imagery is represented by the minimum int16 value.

The VIL images in SEVIR are stored as integers in the range 0-255. It is often convenient to work directly with these encoded units (which was done in the main paper). To convert these into units of $kg/m^2$, which are the true units of vertically integrated liquid, apply the following rule:

$$\text{decoded\_vil} = \begin{cases} 0 & \text{if } X \leq 5 \\ (X-2)/90.66 & \text{if } 5 < X \leq 18 \\ \exp(X - 83.9)/38.9 & \text{if } X > 18 \end{cases} \qquad (2)$$

,

where X is the integer value stored in the HDF file. This non-linear scaling rule was developed to make better use of the range 0-255 for storing VIL. The reason for this is that VIL data exhibits histograms that quite skewed, and the non-linear encoding provides better precision and lessens the skewness. Missing, or `nan` pixels are represented by the value 255 for the `vil` type.

## 2   Model Architecture

The Nowcast and Synthetic Weather Radar models use variants of the U-Net architecture [3]. Figure 1 shows the sizes of the encoding and decoding blocks and the details of each block. Our implemetnation used four encoder blocks with 32, 64, 128 and 256 filters of size 3x3, followed by a bottleneck layer with 1024 `Conv2D` filters with a 3x3 receptive field. Each encoder block consists of two `Conv2D`, `BatchNorm`, `Relu` activation, and finalized with a 2x2 `MaxPool2d` layer. The decoder portion of the network consists of 4 decoder blocks for nowcasting, and 5 in the case of synthetic radar in order to match the target resolution. Each decoder block consists of a `Conv2DTranspose` followed by skip connection with the parallel encoder block, followed by two `Conv2D`, `BatchNorm`, `Relu` sequences. The final layer of the network is `Conv2D` with linear activation, configured with the appropriate number of outputs depending on the application. The final layer of the network is `Conv2D` with linear activation, configured with the appropriate number of outputs.

Figure 1: U-Net model architecture used in this paper: This example shows the Nowcast workflow where the inputs consist of an hour of weather in 13 time steps and the output is the predicted weather for the next hour as represented in 12 images.

## 3   Model Training

All models were trained at the MIT Supercloud high-performance computing system. The compute nodes had two 32GB NVIDIA Volta V100 GPUs and dual 2.5 GHZ Intel Xeon Gold 6130 processor with 20-cores per CPU and a total 384GB of system RAM. Models were trained on a total of 8 GPUs across 4 compute nodes using data distributed training implemented using Horovod [4].

Training effective deep models requires the tuning of hyperparamters which includes learning rates, batch sizes, number of encoder and decoder layers, number of filters per layer, filter sizes per layer, and many other configurable model parameters. A comprehensive analysis of all possible combinations of these parameters is not possible in any reasonable amount of time and we used best practices from prior work to inform our choices. However, the choice of the batch size was dictated by the data sizes used in our study and the loss function used to implement the model. This was particulary important in the Nowcast problem, where the output data sizes are significantly large. Using large batch sizes would lead to out-of-memory (OOM) errors on the GPU and so our choice of

Table 3: Batch sizes and training epochs used for training Nowcast models: The batch size listed in the table is per GPU. Since models were trained across 8 GPUs, the aggregate batch size is 8 times the numbers listed here.

| Nowcast Loss | Batch Size | Training Epochs | Time per epoch |
|---|---|---|---|
| MSE | 32 | 100 | 227 sec. |
| Style & Content loss | 4 | 100 | 2467 sec. |
| MSE + Style & Content loss | 4 | 100 | 3052 sec. |
| Adversarial loss | 32 | 300 | 233 sec. |

Figure 2: Training loss for Nowcast model

batch size was the largest possible for the successful training of the model. The batch sizes used for Nowcast models with different loss functions are listed in Table 3. For the Nowcast application, when using the MSE or adversarial loss, the largest batch size possible per GPU was found to be 32. If data is converted to `float16` data type, a batch size of 48 can be also used in these cases. When the style and content loss was used, the maximum batch size possible was 4. This was due to the fact that the memory usage on the GPU increases significantly due to the use of a VGG network for calculating style and content matrices. The increased memory requirement comes from the requirement that the VGG network is trained on RGB images. Thus, each of the 12 single-channel images in the Nowcast model output must be converted into a three channel pseudo-RGB image. The same operation must be repeated for the truth data, leading to a tripling of the data used to calculate the loss at the end of each batch. This pseudo-RGB data is then passed through the VGG network before the Style and Content features are calculated. The style features consist of the Gram matrices of the output of `block5_conv1` of VGG16 and the content features is the output of `block5_conv2`. The final loss is the mean squared error between the style and Gram matrices of the truth and predictions.

## 4 Metrics

Meterology domain specific metrics for evaluating the performance of the Nowcast and Synthetic Radar applications were calculated on binarized truth and predictions from the models. Figure 4 illustrates one example of a series of `vil` images from SEVIR binarized at different pixel levels. Thresholds for `vil` were chosen based on the 6 Video Integrator and Processor (VIP) intensity levels [2] which correspond to pixel values $[16, 74, 133, 160, 181, 219]$. Binarized pixels are scored as "Hits" if `prediction=truth=1`, "Misses" if `prediction=0,truth=1`, "False Alarms" if `prediction=1,truth=0` and "Correct Rejection" otherwise. The following summary statistics are computed by aggregating these counts over the test set:

Probability of detection (POD) $= \frac{\#Hits}{\#Hits + \#Misses}$

Success Ratio (SUCR) $= \frac{\#Hits}{\#Hits + \#F.Alarm}$

Critical Success Index (CSI) $= \frac{\#Hits}{\#Hits + \#Misses + \#F.Alarms}$

BIAS $= \frac{\#Hits + \#F.Alarms}{\#Hits + \#Misses}$

In the Nowcast applciation, metrics were averages over the 12 steps in the output. Metrics reported in the paper were calculated on an independent valiation set consisting of 12,144 `vil` sequences.

Figure 3: Training loss for Nowcast model with GAN loss

Figure 4: Example of thresholding process used to calculate metrics: Figure shows 12 time steps from the Nowcast dataset created using SEVIR and the output of thresholding the image at each time step. Each row represents thresholded output for a unique threshold value. The binarized images also illustrate the motion in the images.