[Reviews · NeurIPS 2020]

Review 1

Summary and Contributions: The authors present a publicly available dataset for meteorology (SEVIR) containing ten thousand of temporally and spatially aligned images of weather events from different sensing modalities. They also include baseline model implementations, error/loss functions and task applications showing the utility of their data.

Strengths: Generally, the paper is well introduced and easily understood (except for the numerous acronyms). Especially, the provided dataset is very useful in promoting the development of the problem addressed: weather modelling for a range of problems in meteorology (forecasting, img2img translations, caption generation, super-resolution, etc.). Although not being a really “novel” idea/data, providing new benchmarks/datasets/competitions for the AI community is always refreshing. The authors also include a couple of benchmark problems (short-term forecasting and synthetic weather radar generation) and with working implementations of the pretrained baselines and easy-to-understand evaluation metrics allowing the community to keep on working on the task at hand. Illustrative experimental setting showing the usefulness of the datasets and its easiness of use.

Weaknesses: Authors use a very limited number of learning models for evaluation with limited explanations. For instance, differences between those more proficient and worst models should be further studied/explained, although some lines are included by the authors in this regard. In general, although the paper is a very nice engineering effort, it needs more justifications about the new indicators/metrics introduced, the experimental setting proposed (baselines/tasks), and regarding the results obtained in the two tasks addressed (there are not important findings as it is just an application of a few methods…). More effort should be done for a more careful interpretation of what is done and why. The conclusions are a summary and I would expect (again) more insight and future work.

Correctness: The experimental setting follows the standard ML evaluation methodologies.

Clarity: Well written except for the numerous accronyms used.

Relation to Prior Work: Good.

Reproducibility: Yes

Additional Feedback:


Review 2

Summary and Contributions: This paper introduces a new dataset which combines and aligns multiple modalities from land-based meteorological radar and geostationary weather satellites. Additionally, they select a large subset of the dataset based on a storm event database and provide a large amount of additional metadata. On the given dataset, authors evaluate several baselines, where quite a few of them are novel. To the best of my knowledge, there is not an existing literature with training nowcasting predictors with conditional GAN and style losses.

Strengths: The selected dataset is really interesting and in my opinion would be really useful for the community. Authors also provide a wide array of possible tasks from the dataset and show results on two of them, where the second task is about predicting different modalities from satellite data which I find really interesting. I can imagine the array of possible experiments can be even wider. The baselines by themselves are quite novel and authors correctly point out that pixelwise losses are problematic as they provide blurred predictions. Authors provide full source code of all models.

Weaknesses: It would be useful to specify a validation set which would help careful researchers to avoid possible over-fitting to the test set and would allow use of early stopping etc. Additionally, more existing baselines, mainly a Lagrangian Persistence (Optical flow), which is one of the main methods used in practice, and quite a few implementations are part of open source projects. Additionally, ConvLSTM and TrajGRU models are also really important, but probably not completely necessary. Additional issues are more minor. I think the manuscript should clearly state that using the pre-trained VGG network for anything (both for losses and metrics) should be seen only as proof of concept. It is not at all clear that features trained on natural image classification are going to be anyhow related to this completely different domain of data. However, this is an open research topic. When it comes to evaluation metrics, for readers of this conference it would be useful to mention the standard nomenclature, such as HITS=TP, FAlarms=FP, POD=recall etc. For each metric it should be stated for which higher is better. It appears that the bias is misrepresented by the authors as lower is better (as BIAS indicates whether model is playing it safe and over-predicts the precipitation). With regards to the binary threshold, it would be much better to report the metrics against the more common mm/h which is used in almost all other literature and does not depend on a specific location or algorithm used for obtaining the radar mosaic. It would be also an interesting to see what is the influence of the non-linear VIL transformation (described in the supplementary material) on the training results. Also, it might be useful to mention the license of the dataset.

Correctness: Yes, the method appears to be correct.

Clarity: Yes, this paper is well written.

Relation to Prior Work: This paper introduces a new dataset, and evaluated main baselines based on previous works.

Reproducibility: Yes

Additional Feedback: [Post rebuttal] I would like to thank the authors for the thorough rebuttal. I was really surprised to see the optical flow results, as it strengthens the submission even more! However, it would be important in the final submission to address the fact that optical flow performs so well on the LPIPS scores (which is quite interesting by itself). With regards to the units, I agree that simplicity of the pipeline is important. However at least a conversion table in the supplementary materials and a mention of physical units for the selected 74/133 binary metrics thresholds would go a long way, as it would provide interpretability to a practitioner with a forecasting background.


Review 3

Summary and Contributions: The paper contributes a large-scale multi-modal dataset of severe weather events called SEVIR and proposes a number of machine learning challenges and opportunities. Data from multiple sources (satellite imagery, RADAR) are spatio-temporarily aligned and are provided with labels obtained from NOAA's storm event's database resulting in nearly a petabyte of data. Experiments in the paper suggest that the data is sufficient to train a ML model for weather nowcasting that beats baselines.

Strengths: Creating a high quality dataset is challenging. The paper comes across as a significant and thoughtful effort by domain experts. Such datasets are urgently lacking to train and evaluate ML / CV models. The paper is highly relevant as Earth-scale data is increasingly becoming available. However, a barrier is that accessing them requires domain expertise and engineering skills that most ML experts don't have. The paper takes an important step in this direction by curating a dataset and providing a way to access it in a ML-ready format. The paper also articulates several tasks and evaluation metrics that are relevant to the weather prediction community. The baseline experiments and loss functions used for training are well executed. In summary, the contributions of the paper is a novel dataset.

Weaknesses: Perhaps the main drawback of the paper is that the method and theoretical contributions are weak. While the novelty of the approach isn't required (or the primary concern), the evaluation could be improved and should be presented in the context of existing work in the weather prediction community. For example, the results are compared against a baseline ("no change") model. What other approaches are there for this problem and does ML provide an improvement? As a result it is hard to judge the significance of the method (e.g., will these models be adopted in the community?).

Correctness: The paper appears to be correct. The methods are based on standard CV models for image segmentation and image to image translation. The loss function for training is primarily investigated.

Clarity: The paper is well written and easy to follow.

Relation to Prior Work: The paper is primarily a dataset paper. However, a discussion of related techniques for the proposed problem is missing.

Reproducibility: Yes

Additional Feedback: The paper is likely outside the expertise of most NeuRIPS audience. Some more background on dataset creation and the value of a ML solution would make the introduction more compelling. The reviewer has some prior experience on working with NEXRAD data, yet I'm not aware of some of the data sources (e.g., GOES.) ### Post rebuttal comments ### Appreciate the optical flow baselines in the rebuttal and answers to the issues surrounding metrics and benchmarks. Incorporating these in the main paper will improve the accessibility to a broader ML audience.


Review 4

Summary and Contributions: [POST REBUTTAL] I think authors discussed main reviewers points in the feedback (some very interesting), and I am convinced that changes in the paper can only improve the first submission. For this reason, I raised my score from 6 to 7, and I hope that the interest of the ML community to Earth sciences keep growing. I agree in general with Authors that adding baselines would be a very interesting thing, but probably their evaluation requiring a separate paper with more domain specific evaluations. From my side, my comments were asking for more details and (short) discussions, which are addressed in the feedback and authors will add some of them in the final version. The paper presents a large and multimodal dataset for meteorology applications, involving weather events. The paper mentions few potential applications and studies two particular ones, providing deep learning baselines. Authors mention that they will release code and data readers, models and everything needed to reproduce and extend the work.

Strengths: + The dataset is indeed new and potentially of interest. It lies at the intersection of the ML and meteo / weather communities, and poses non trivial problems for both wolds. + Releasing data readers and the metadata required would allow a seamless expansion of the dataset, including new modalities and new events.

Weaknesses: - It is not really clear what types of event are available, and what is the difference between events and episodes. - It is clear the usefulness of the tested baseline in terms of nowcasting, but I wonder if another very useful application (at least for insurance and archiving) would be the classification of events. This goes with the point above, it would be nice to have a more in-depth description of the events, their frequency and the possibility of predicting them given the input. would it be only a system based on thresholds on averages of the measures, or ML would be really required? - Following the same line, I was wondering what is the spatial accuracy of the events. Is it simply a label attached to the 384^2 km inputs or is it localized within each image, for each time? what is the role of the clustering and how sensitive is the aggregation to the clustering architecture? - Is the choice of the mean errors as loss correct? It seem that the values of weather radars are very skewed towards 0s, and large values very rare. this would make the problem maybe closer to ordinal classification? Also, I wonder if maybe there are some more domain specific loss functions to be optimized, eg taking into account spatial smoothness of signals, rarity of levels, level sets of precipitation, etc. I understand testing everything out is not posisble, but maybe a discussion is welcome by readers of both ML / meteo communities. - I think that a very useful addition would be to describe in more detail: -- The transferability of the models to other countries and continents (I think further work would be needed, since weather patterns, their classification, their modalities might differ heavily over different countries / regions). Is the CONUS showing enough variation to be able to predict weather patterns over other areas of the Earth? -- The quality of the results. I understand that a lenghty discussion about the quality of the results is maybe not easy to fit into the paper, but I would really love to see a short discussion about how realistic predictions are, from a meteorological and weather science perspective, given the different losses. Why one loss should be better from the other from a domain science perspective? This would give a strong signal to the fact that ML can indeed help weather sciences. -- A discussion on limitations of the model used, in light of the points above. I keep saying "discussion", but it can be a very short paragraph / couple sentences. I think that this is important to address the reader and to give credibility to the dataset vs current ML methodologies. e.g. to avoid the "I use a big deep net on every problem of section 3 and I get 99% accuracy and therefore realistic models" type of thinking (which I think is false in many cases, even more so here).

Correctness: The paper mostly presents a dataset and two applications of deep learning models. From a deep learning model perspective, authors perform direct transfer from computer vision tasks / losses, without expanding and considering maybe more data specific approaches. Still, this might work, but that is why I asked above for a discussion on how results are realistic.

Clarity: Yes, the paper is clear and well written.

Relation to Prior Work: Prior work is probably scarce, but authors reference where needed.

Reproducibility: Yes

Additional Feedback: I think the paper is good overall. The only weak point I can imagine is that is maybe too application oriented for NeurIPS, but as there will be surely an application track or environmental sciences track, I think it can fit the conference. I miss overall some more discussions on the correctness of results from a domain science perspective, and potentially a more in depth discussion on the limitations of current ML models to address these problems.

[Author Response · NeurIPS 2020]

Thank you to each reviewer for your helpful feedback on our paper. All comments were taken under consideration and will be used to improve the work. Below we provide our reasoning for several selected points.

**Lack of novelty / Use of existing methods** While it might come off as an engineering task, we feel the design and execution of SEVIR required several novel ideas and insights, including recognition of a gap in ML-ready weather datasets, application of the NOAA storm event database for down selection, combining/structuring/formatting multiple data sources together in a way that makes it relevant to a number of tasks in the intersection of meteorology and machine learning. Also, the use of existing methods and metrics in the second half of the paper was an intentional choice by the authors; the goals of the paper are to introduce a new dataset to the community, and to provide simple baselines that are standard, easy to understand, and are reproducible. Inclusion of more complicated or novel architectures/methods would have put us beyond that scope, and moreover would have been unfair to the researchers developing those methods since those results are worthy of their own paper on the topic. Contrary to prior work in this area, we evaluate our models using not only meteorology specific metrics but also with metrics that quantify perceptual quality. This also highlights the fact that a single metric is not always sufficient for evaluating models where image quality perceived by the viewer is also important.

**Lack of Benchmarks** Multiple reviewers commented on the weakness of the benchmark used for the nowcasting task, and Reviewer 2 was correct to point out several existing nowcasting methods that are in use that go beyond the "do nothing" persistence model. The choice of only using the persistence benchmark came down to two main reasons (1) Despite it's simplicity, it is a common benchmark used for nowcasting and (2) It is hyperparameter free – unlike other methods for this task (e.g. optical flow). Having said that, based on this reviewer feedback, we have included an optical-flow based benchmark using on SEVIR and intend to include it as a additional baseline if the paper is accepted for publication. Due to page limits, only a portion of the updated figure is shown below. Since we want the focus of the paper to be the utility of the dataset for AI/ML in meteorological applications, we decided an in depth and fair comparison to additional methods (e.g. TrajGRU) would be out of scope (and well over page count). Additionally, evaluation, hyperparameter and architectural tuning of ConvLSTM is part of an ongoing effort and an in-depth comparison of Nowcast models is currently under way.

**Discussion surrounding results, metrics and future work** Several reviewers noted weak discussion surrounding metrics and evaluation of the predictions in context of meteorology. This is a fair criticism, and thankfully, is something that could be remedied in the final version (given the extra page alloted). Evaluation metrics for weather prediction is a rich area since the task is multi-objective – in some cases the placement and intensity of weather is most important, and in others the realism and dynamics of the storm are more important. This motivated the range of different losses and objectives, some that stress placement/intensity (MSE/POD/CSI), and others that focus on realism (GAN-based/LPIP). The baselines we provide show that depending on your choice of loss function, certain axes of "goodness" are brought out more than others. We will add more discussion along these lines which address "what is done and why". Also, Reviewer 1 pointed out, our conclusion is simply a summary. We can use the extra space to discuss future and ongoing work, which include a planned AI challenge in the area of radar/satellite nowcasting, applications of SEVIR to few-shot learning that aims to address one of Reviewer 4's questions ("The transferability of the models..."), and approaches that improve robustness in ML models applied to weather.

**Clarity / Use of acronyms** For evaluation metrics, we used naming conventions commonly used in the meteorological community (POD vs recall, FAR vs precision, CSI vs IOU). In hindsight, perhaps that wasn't a good choice for NeurIPS. Following Reviewer 2's suggestion, the acronym problem can be alleviated by switching to more standard terminology.

*To Reviewer 2*: Agreed about VGG, it's actually surprising how well VGG works in this setting despite being trained on completely different data! Also, we can provide metrics in the actual units of the fields. Our reason for using the 0-255 version of radar was to simplify the implementation by not requiring additional post processing, but that's easy to add.

*To Reviewer 4*: You're right to point out the word "event" is a bit overloaded, but the NOAA storm events (which are clustered into "episodes") can be linked to particular pixels and frames of the SEVIR events using data in the catalog (so NOT the whole $384^2$ km region). Regarding loss functions: this is very much an open question. Most work uses some $L^p$ loss as it's the simplest, but to your point, there may be better ones. We started to explore that question here with the adversarial and texture losses. Our vision is to spur innovation in this area not only through model architectures, but also new loss functions that are able to simultaneously model weather placement/intensity as well as realism, through a future AI challenge that leverages SEVIR and these baseline implementations.

[Meta-Review · NeurIPS 2020]

Four knowledgeable reviewers all appreciated the contributions of this paper and rated it as above the bar for publication at NeurIPS. Reviewers acknowledged that the primary contribution was the curation of a dataset and benchmark tasks on the data set, and not novel methods, but felt that the curation of a large, high-quality data set for real tasks in atmospheric/earth sciences is important and could spur AI work in this area. The authors deserve credit for this. Additionally, the reviewers appreciated that the baseline methods developed for the benchmark tasks were themselves thoughtful and significant, if not highly novel from a methods perspective. The reviewers asked a number of questions about justification and details of the data set construction, evaluation metrics, and baselines. The authors answered several of these in the rebuttal, including adding a new baseline method (optical flow); they are encouraged to use these comments to improve the final version of the paper. The meta-reviewer recommends accept.